# Structure and Anticoagulant Activity of a Galactofuranose-Containing Sulfated Polysaccharide from the Green Seaweed, *Codium isthmocladum*

**DOI:** 10.3390/molecules27228012

**Published:** 2022-11-18

**Authors:** Peipei Li, Junlu Bai, XiaoJun Zhang, Zhongyong Yan, Pengfei He, Yin Chen

**Affiliations:** 1Zhejiang Provincial Key Lab of Mariculture & Enhancement, Zhejiang Marine Fisheries Research Institute, Tiyu Road 28, Zhoushan 316021, China; 2College of Food and Pharmacy, Zhejiang Ocean University, 1 South Haida Road, Zhoushan 316000, China

**Keywords:** marine green alga, polysaccharide, galactofuranose, anticoagulant activity

## Abstract

A water-soluble sulfated polysaccharide, F2-1, was obtained from the marine green alga, *Codium isthmocladum*, using ion-exchange and size-exclusion chromatography. Structure analysis showed that the F2-1 was a sulfated arabinan comprising Ara, Rha, Man, Gal, and Xyl with an 18% sulfate content and a molecular weight of 100 kDa. Methylation analysis combined with desulfation, GC-MS, IR, and NMR spectroscopy showed that the backbone of F2-1 was →4)-β-L-Ara*p*(1→ residue. Its 2-*O* and/or 3-*O* positions showed sulfate modification; additionally, the 2-*O* or 3-*O* position showed branch points. The side chains were composed of →5)-β-D-Gal*f*, (1→2,6)-β-D-Gal*f*(1→, (1→2)-β-L-Rha*p*4S, →4)-α-D-Glc*p*(1→, and terminal α-D-Gal*p*(1→ and β-D-Xyl*p*(1→. Polysaccharides containing β-D-galactofuranose are rarely found in seaweed. F2-1 exhibited significant anticoagulant activity in vitro. Our findings suggested that the green-tide alga, *Codium isthmocladum*, can be considered as a useful resource for bioactive polysaccharides.

## 1. Introduction

Green seaweed has been used as natural materials for the extraction of bioactive substances for over the past 20 years, owing to their widespread distribution and large biomass [1]. They are usually grown and collected for consumption as food and are especially known for their high nutritional value and health benefits. However, among the three main divisions of macroalgae (i.e., Chlorophyta, Phaeophyta, and Rhodophyta), marine green algae remain largely unexploited [2]. There has been a recent increased interest in using green seaweeds as a natural resource owing to their multiple active ingredients, including bioactive lipids, proteins, peptides, polysaccharide, carotenoids, phenols, alkaloids, and others, that could potentially be used for medicinal purposes [3]. Among these active ingredients, polysaccharides as rich and important sources of bioactive natural polymers have been extensively investigated for medicinal purposes [4,5,6,7,8]. The polysaccharides from green seaweeds have a wide range of physiological and biological activities, including immunomodulatory, anti-inflammatory, antioxidant, anticoagulant, and antitumor effects [2,9,10,11]. Sulfated polysaccharides (SPs) from green seaweeds are chemically and physicochemically different [12,13], and may have various physiological effects on the human body. Ulvans and their oligosaccharides are well-known SPs obtained from the green seaweed Ulva and Enteromorpha. Sulfated galactan and arabinan represent another kind of polysaccharide from the green seaweed *codium*. These compounds have demonstrated strong antitumor, immunomodulatory, antihyperlipidemic, and anticoagulant activities [7,11,14]. Considering these biological activities, the extraction of polysaccharides from green seaweed and their use as therapeutic agents are becoming increasingly important topics of research. 

The development of algal blooms (principally *Gracilaria tikvahiae*, *Laurencia poitei*, and *Codium isthmocladum*) along the south-eastern coast of Florida, USA was the subject of studies for more than a decade and a half [15]. Its adverse environmental impact was periodically and seasonally evident. Owing to decaying corals and the presence of macroalgae, water quality near the shore has deteriorated and tourism has been affected by the declining “quality” of the beaches. In several countries’ “green tides”, algae are harvested to play a role in pollution-abatement scenarios [16]. Algae play a significant role in pollution abatement and could potentially be used for energy production. They have been used in the past for electricity generation [17]. The research by Bellan [18] showed that sulfated homogalactan (named 3G4S) from *Codium isthmocladum* seaweed was able to reduce solid tumor growth and metastasis while not inducing side effects in mice, and could impair B16-F10 cells traits related to the metastatic cascade, reducing cell invasion, the colony-forming capacity and membrane glycoconjugates. Therefore, 3G4S shows promising antitumor activities without the commonly associated drawbacks of cancer treatments. Farias [19] studied the structure of sulfated galactan from *Codium isthmocladum*. This sulfated galactan was composed preponderantly of 4-sulfated, 3-linked β-D-galactopyranosyl units with a sulfate group located at C-6.

In this study, polysaccharides from the green tide seaweed, *C. isthmocladum*, were studied to derive more ideas for the optimal utilization of this seaweed to efficiently solve the growing problem of global pollution [20].

## 2. Results and Discussions

### 2.1. Extraction and Characterization of the Sulfated Polysaccharides 

The water-soluble polysaccharides extracted from *C. isthmocladum* were obtained as a total yield of 27% of dry algae powder. Using Q Sepharose Fast Flow column chromatography, the polysaccharides were separated into distilled water, 0.5, 1, and 2 mol/L NaCl-eluted fractions (Figure 1a). The 0.5 mol/L NaCl-eluted fraction, F2, was the major component. As it had the best solubility and anticoagulant activity, F2 was chosen for further studies and purified using a Superdex 75 column. The major fraction was collected based on detection using the phenol-H_2_SO_4_ method and named as F2-1 (Figure 1b).

Colorimetric assays indicated that F2-1 had a high carbohydrate (73%) and sulfate (18%) content and only trace amounts of uronic acids; pyruvic acid ketals were not detected. The protein content of this extract was determined to be 6.2%. HPGPC analysis revealed a symmetric peak of F2-1, which indicated that it was a relatively homogeneous polysaccharide (Figure 1c). Based on the standard calibration curve, its molecular weight was estimated to be 100 kDa. Monosaccharide composition analysis showed that F2-1 had the highest levels of Gal and Rha. Glc, Ara, a low level of Man, and trace amounts of Xyl were also detected in F2-1 (Figure 1d). The molar ratio of these monosaccharides Man:Rha:Glc:Gal:Ara was 1.0:4.0:2.5:6.8:2.8.

### 2.2. Infrared Spectra of F2-1 and Its Desulfated and Methylated Derivatives

The FT-IR spectrum of the sulfated polysaccharide F2-1 is shown in Figure 2. The broad and intense signal at 3247 cm^−1^ is characteristic of the stretching vibration bend of the O–H of sugar. The peaks at 2940 and 1421 cm^−1^ can be attributed to the stretching vibration and deformation vibration of C–H. Considering the absence of uronic acids in F2-1, the intensive band on the 1646 cm^−1^ could not be the peak of the C=O bond, but corresponds to bound water of the polysaccharides. The band at around 1051 cm^−1^ has been assigned to the stretching vibrations of the C–O–C bond and glycosidic bond, whereas that at 1250 cm^−1^ has been assigned to stretching vibrations of S=O. The signals at 845 cm^−1^ can be attributed to the bending vibration of the C–O–S of the sulfate ester in the axial position. These two peaks also indicated the existence of the sulfate group in F2-1. The completion of methylation was confirmed by the disappearance of the O–H signal at 3247 cm^−1^ and an intense increase in the CH_3_ signal at 2940 and 1421 cm^−1^. Desulfation was confirmed by the disappearance of the C–O–S and S=O bands at 1250 cm^−1^ and 845 cm^−1^, respectively.

### 2.3. Methylation Analysis of F2-1 and Its Desulfated Derivative

We used the classic method of methylation analysis to obtain structural information of the sulfated polysaccharide (F2-1) and its desulfated derivative (DsF2-1) (Table 1). Ara in F2-1 had the linkages of terminal Ara(1→, →4)Ara(1→ and →2,3,4)Ara(1→. After desulfation, the ratio of →4)Ara(1→ increased obviously, whereas the linkage of →2,3,4)Ara(1→ decreased drastically. These results indicated that Ara in F2-1 mainly had →4)Ara(1→ linkage. Sulfate groups were substituted at the 2, 3-O positions of →4)Ara(1→. Similar to that of Ara, an increase in →2)Rha(1→ and decrease in →2,3)Rha(1→ and →2,4)Rha(1→ also suggested that Rha in F2-1 existed with the →2)Rha(1→ linkage and had sulfate substitution at the 3-O or 4-O positions. The presence of 1,4,5-tri-O-acetyl-2,3,6-tri-O-methyl-D-Gal, 1,2,4,6-tetra-O-acetyl-3,5-di-O-methyl-D-Gal, and 1,5-di-O-acetyl-2,3,4,6-tetra-O-methyl-D-Gal in F2-1 indicated that the Gal units were in both furanose and pyranose forms. These furanosic galactose units (→5)Gal*f*(1→ and →2,6)Gal*f*(1→) were present only in minor amounts in DsF2-1, indicating that they were lost during desulfation. The glycoside made from Gal*f* was highly sensitive to pH. The furanosic glycosidic linkages were degraded during the preparation of the pyridinium salt of the sample (pH 3.5) even at room temperature [21]. In addition to these linkages, F2-1 also contained terminal Man*p*(1→, Glc*p*(1→ and →4)Glc*p*(1→ linkages.

### 2.4. NMR Spectroscopy of F2-1

The ^13^C-NMR spectrum of F2-1 shows eight signals in the anomeric region at 97–108 (Figure 3b). The 1H-NMR spectrum shows a significant overlap of the anomeric signals (Figure 3a). The other signals of the protons were in the region of 3.0–4.3 ppm. The signal at 1.2 ppm could be that of the -CH_3_ of Rha. The anomeric region of the HSQC spectrum of F2-1 (Figure 3e) shows eight anomeric signals in the 5.3–4.4 ppm region that are designated as **A**–**H** based on the decreasing chemical shifts of their anomeric protons. The two major signals at **C** (δ 5.16/108.8) and **E** (δ 5.0/107) are attributed to the anomeric carbons of the β-galactofuranose units, which are owing to their low-field resonances. Even the anmeric carbon of Ara*f* was at the same region with galactofuranose; these two signals were assigned as Gal*f* based on the data of GC-MS. The complete assignment of the residues was achieved using H-H COSY and TOCSY (Figure 3c,d). **C** was deduced to be →5)-β-D-Gal*f*(1→ because of the chemical shift of C5 downfield at 76.1 ppm [22]. The extreme downfield shifts of C2 and C6 of E at 83.7 and 69 ppm also confirmed that E had 2-*O* and 6-*O* substitutions. These observations revealed E to be (1→2,6)-β-D-Gal*f*. Compared with the references [23], **D** (5.04/98.3) represented the signal of →4)-β-L-Arb(1→ with no sulfate substitution. **A** (5.31/99) represented (1→4)-linked 2,3-sulfate-β-L-Ara*p*. Sulfation on C2 should produce a downfield shift of the anomeric proton. Moreover, the H1 signal at 5.31 correlating with H2 at 4.7 in the COSY spectrum also indicated 2-*O* sulfate substitution, whereas H3 at 4.5 ppm indicated 3-*O* sulfate substitution. These correlations and determinations allowed for the assignment of these residues. Two anomeric protons at the upfield (**G** 4.57 and **H** 4.46 ppm) were correlated to C1 at 103.3 and 103.6 ppm, respectively. **G** was assigned to β-L-Rha units because of the C6 at 1.2 ppm. The chemical shifts of H2 and H4 (4.3 and 4.04 ppm, respectively) and the methylation results indicated that **G** was (1→2)-linked 4-sulfate-β-L-Rha*p*. **H** was a terminal β-D-Xyl*p* unit. Residue **C** (5.2/97.4) with H2 at 3.55 ppm and C4 at 76 ppm could be →4)-α-D-Glc*p*(1→, whereas **F** could be terminal α-D-Gal*p*(1→ unit [24]. The chemical shifts of carbon and hydrogen for major residues are listed in Table 2.

The correlation signal of **C** H1/C2 **E** indicated that →5)-β-D-Gal*f*(1→ was linked to the *O*-2 position of →2,6)-β-D-Gal*f*(1→. Owing to the complexity of the polysaccharide structure and the lack of relevant signals, it was a challenge to analyze the HMBC spectrum to determine how the galactofuranose units were linked to the Ara backbone. These results, combined, show the structural features of the sulfated, branched heteropolysaccharide, F2-1 (Figure 4). F2-1 shows →4)-β-L-Ara*p*(1→ as the main chain and 2-*O* and/or 3-*O* to have sulfate substitutions. Meanwhile, the 2-*O* or 3-*O* positions were the branch points. The side chains were composed of →5)-β-D-Gal*f*(1→, →2,6)-β-D-Gal*f*(1→, →2)-α-L-Rha*p*(1→, →4)-α-D-Glc*p*(1→, terminal α-D-Gal*p*(1→, and β-D-Xyl*p*(1→. The 4-O position of →2)-β-L-Rha*p*(1→ had sulfate substitution.

Sulfated arabinans and arabinogalactans with pyranose configurations have been mainly found in green seaweed belonging to the genus *Codium* [25,26]. However, most of these arabinans are linear polysaccharides with →3)-β-L-Arb*p*(1→ as the backbone, with sulfate group substitutions at the 2-*O* or 4-*O* positions. →4)-β-L-Arb*p*(1→ with partial sulfate modification on C3 has also been reported in the green seaweed *Caulerpa racemosa* (Bryopsidales) [27,28,29,30]. Another interesting finding was that Gal in F2-1 had both pyranose and furanose rings. Polysaccharides containing β-D-galactofuranose are very rare in plants or animals. Galactofuranose-containing polymers have been found in various microorganisms, such as fungi, bacteria, and lichen; however, they have not been reported to exist in mammals [31]. Additionally, they have not been found in any other seaweed except for some green microalgal species involved in a mutualistic relationship with terrestrial fungi (lichens) [32].

Sulfated galactan, which was composed preponderantly of 4-sulfated, 3-linked β-D-galactopyranosyl units, was found in *C. isthmocladum* in braizle. Ara and Man were also found in other polysaccharide fractions of this species. However, the structures of these heteropolysaccharides were not elucidated. In this paper, we found a very different poly-

Saccharide that contained galactofuranose from the same species. Our finding expanded the polysaccharide composition of this species [19].

Galactofuranose with a high structural diversity has been proposed as a chemotaxonomic character. A water-soluble, sulfated xylogalactoarabinan from the green seaweed, *Cladophora falklandica*, shows similar structural characteristics. The xylogalactoarabinans also have a backbone of four linked β-L-arabinopyranose units partially sulfated mainly on the *O*-3 and *O*-2 positions. In addition, they have a single terminal β-D-galactofuranose or →5)-β-D-Gal*f*(1→ and →6)-β-D-Gal*f*(1→ in the side chains. Compared to this xylogalactoarabinan, F2-1 differed in that it had more Rha and less terminal β-D-galactofuranose in the side chains [23,33]. F2-1 had →2,6)-β-D-Gal*f*(1→ but not →6)-β-D-Gal*f*(1→ residue. These findings suggested that some species in green seaweed could also produce galactofuranose-containing polymers. This seaweed may have acquired genes from microorganisms. The study of this aspect could serve as an interesting topic for future research. Ara was found to be a constituent of arabinogalactans from the green seaweed *Caulerpa racemosa* (Bryopsidales) and *Cladophora rupestris* (Cladophorales). In addition, the water-soluble polysaccharides from *Cladophora falklandica* comprised Gal units in the furanose form. Similar findings were observed in our current study. Thus, this is the second instance where galactofuranose has been discovered among seaweed polysaccharides. Collectively, these results showed that the presence of sulfated arabinans and galactofuranose side chains is a characteristic of green algae belonging to Cladophorales and Bryopsidales.

### 2.5. Anticoagulant Activities

Anticoagulant activity has been reported as a classic activity of sulfated polysaccharides. To explore and utilize the polysaccharide resource in waste, green algae F2-1 was screened to determine its anticoagulant activity using APTT, TT, and PT assays, with heparin as a control. As seen in Table 3, F2-1 could effectively prolong APTT and TT. The clotting time of F2-1 was more than 200 s for APTT and 120 s for TT at a concentration of 20 μg/mL. F2-1 had no effect on PT. The coagulation process includes the intrinsic and extrinsic pathways, and APTT and PT are related to the intrinsic and extrinsic pathways, respectively. TT indicated the function of thrombin activity and fibrin polymerization. For F2-1, a lack of PT-prolongation activity demonstrated no inhibition of the extrinsic pathway of coagulation. Thus, it was presumed that F2-1 acted on the intrinsic pathways of coagulation and thrombin activity or in the conversion of fibrinogen to fibrin. APTT and TT activities are rapidly increased by heparin. Heparin exerts anticoagulant effects indirectly by binding with antithrombin III (AT) and facilitating the subsequent inhibitory effect of AT on thrombin and activated factor X (factor Xa). Xa is the key factor present in both intrinsic and extrinsic pathways; therefore, heparin extended the clotting time in all assays [34]. However, in F2-1, it might have affected the factors present only in the intrinsic pathways. Therefore, compared to heparin, F2-1 had mild anticoagulant activities.

Green seaweed sulfated polysaccharides used as anticoagulants are obtained mainly from species of *Codium* because sulfated polysaccharides isolated from *Codium* seaweeds exhibit remarkably potent activity. Shanmugam [35] further investigated the anticoagulant activity of sulfated polysaccharides from 13 species of *Codium* collected from the Indian coast. The results revealed that several *Codium* species, such as *C. dwarkense*, *C. indicum*, *C. tomentosum*, and *C. geppi*, contain polysaccharides with strong anticoagulant activity. The anticoagulation mechanism of these polysaccharides is attributed to the direct inhibition of thrombin and potentiation of antithrombin III. Our study revealed that the galactofuranose-containing arabinogalactan in *C. isthmocladum* also exhibited strong anticoagulant activity by a prolonged activated partial thromboplastin time (APTT), but not through the prothrombin time (PT). Thus the anticoagulation mechanism might be a little different.

## 3. Materials and Methods

### 3.1. Collection of Algal Samples

Specimens of *C. isthmocladum* were collected in the sea around Miami Beach on October 2017. The seaweed was washed with seawater and dried in the sun in the open air. 

### 3.2. Extraction and Purification of the Polysaccharide

The sun-dried alga was pulverized in a blender to yield a powder. Dry algal powder (200 g) was immersed to distilled water (2 L) for 4 h at 80 °C. After filtration, the residue was extracted repetitively 2 times using distilled water. The combined extracts were concentrated to approximately one-tenth the original volume. Polysaccharides were obtained by precipitation with 80% ethanol (final concentration). After centrifugation (15 min at 6000× *g*), the precipitate was dissolved in distilled water. The resulting precipitate was removed by centrifugation and the supernatant was collected. It was dialyzed (molecular weight cutoff MWCO, 3 kDa) against distilled water for 48 h and the crude polysaccharide was obtained as a white powder with a yield of 27% (*w*/*w*, in dry mass) after freeze-drying [21]. The polysaccharide was then purified on a Q Sepharose Fast Flow column (300 mm × 30 mm). Briefly, the crude sample (500 mg) was dissolved in distilled water (50 mL) and centrifuged, and the supernatant was applied to the column (previously stabilized in H_2_O). The sample was first eluted with water, followed by NaCl solutions with increasing concentrations of up to 4 mol/L. The fractions were collected and the content of the carbohydrates in each fraction was determined by phenol–sulfuric-acid method. Four sugar-containing fractions were obtained, dialyzed, and freeze-dried (F1-F4), and F2 was chosen to be further purified on a Superdex 75 column (100 cm × 2 cm) using 0.2 mol/L NH_4_HCO_3_ at a flow rate of 0.3 mL/min. The major polysaccharide fractions were pooled, freeze-dried, and named F2-1. The purity and molecular weight were determined using high-performance gel permeation chromatography (HPGPC) as described in a previous study [36].

### 3.3. Chemical Analyses

The polysaccharide, F2-1 (5 mg), was hydrolyzed using 1 mL of 2 mol/L trifluoroacetic acid (TFA) at 105 °C for 6 h. TFA was removed by repeated co-evaporation with methanol and the hydrolysate was dried under reduced pressure. Uronic acid content was estimated by the carbazole–sulfuric-acid method [37]. Pyruvate acid content was determined by dinitrobenzene removal method [38]. The monosaccharide composition was analyzed using HPLC through pre-column derivatization with 1-phenyl-3-methyl-5-pyrazolone (PMP) using an Agilent HPLC system fitted with an Agilent XDB-C_18_ column (4.6 mm × 250 mm) and Agilent XDB-UV detector. The sulfate content of the polysaccharide was determined using high-performance anion-exchange chromatography with pulsed amperometric detection (HPAEC-PAD) on a CIC-100 ion chromatograph coupled with SH-AC-1 anion exchange column (4.6 mm × 250 mm, 13 µm) and conductivity detector [39]. The total sugar content was analyzed using the phenol–sulfuric-acid method using galactose as a standard [40], and protein content was measured using the Lowry method [41].

### 3.4. Desulfation and Methylation Analysis

To obtain information of the linkages and the positions of the sulfate groups, the sulfated polysaccharide was subjected to desulfation and methylation analysis. The comparison of the gas-chromatography–mass-spectrometric (GC-MS) information of the sulfated polysaccharide, and its desulfation product could provide evidence of the linkages and the sulfate positions.

The desulfation reaction was carried out as described [42]. Briefly, the sample (40 mg) was exchanged from its Na^+^ to H^+^ form using a 732 cation exchange dowex resin column (H^+^ form) and mixed with pyridine to obtain the pyridinium salt. The product was dissolved in DMSO (10 mL) containing 10% (*v*/*v*) anhydrous methanol and 1% pyridine and heated at 100 °C for 4 h to promote desulfation. The effect of methylation and desulfation was confirmed by IR.

Methylation analysis was performed using the method reported by Hakomori, with some modifications [43]. Each sample in dimethyl sulfoxide was methylated using NaH and CH_3_I. The completeness of methylation was confirmed using IR spectroscopy. After hydrolysis with 2 mol/L TFA at 105 °C for 6 h, the methylated sugar residues were converted to partially methylated alditol acetates by reduction with NaBH_4_, followed by acetylation using acetic anhydride. The derivatized sugar residues were extracted into dichloromethane and evaporated to dryness, re-dissolved in 100 μL dichloromethane, and analyzed using GC–MS on DB 225 using a temperature gradient: initially 100–220 °C, with an increase of 5 °C/min, and then held at 220 °C for 15 min [44]. The peaks on the chromatogram were identified based on their retention times. 

### 3.5. NMR

A 600 MHz ^1^H-NMR, proton decoupled 125 MHz ^13^C-NMR spectra, and two-dimensional NMR spectra (HSQC, HMBC, and COSY) were recorded at 23 °C using a JEOL JNM-ECP 600 MHz spectrometer (JEOL, Tokyo, Japan). The samples (60 mg) were exchanged in 99.9% D_2_O (0.8 mL) three times. Chemical shifts were referenced to internal acetone (*δ*H 2.2, *δδ*CH_3_ 31.1). 

### 3.6. Anticoagulant Activity

The in vitro anticoagulant activity was determined based on the prothrombin time (PT), activated partial thromboplastin time (APTT), and thrombin time (TT) assays. For the APTT assay, briefly, 90 μL of citrated normal human plasma and 10 μL of sample solution (0–50 μg/mL) were incubated at 37 °C for 60 s. Then, 100 μL of the prewarmed APTT assay reagent was added and allowed to react at 37 °C for 2 min. Thereafter, 100 μL of 0.25 mol/L pre-warmed calcium chloride was added and APTT was recorded as the time of clot formation. The TT assay was performed as follows: 90 μL of citrated normal human plasma was mixed with 10 μL of polysaccharide solution (0–50 μg/mL) and incubated at 37 °C for 60 s. Then, 200 μL of pre-warmed TT assay reagent (37 °C) was added to the mixture and the clotting time was recorded. For the PT clotting assay, 90 μL of citrated normal human plasma was mixed with 10 μL of polysaccharide solution (0–50 μg/mL) and incubated at 37 °C for 1 min. Then, 200 μL of pre-incubated PT assay reagent (37 °C, 10 min) was added to the mixture and the clotting time was recorded. The clotting assays were performed in triplicate and the results are expressed as mean values ± standard deviations (SD). Heparin was used as a positive control to compare the anticoagulant activity of the fractions, whereas saline (0.9% NaCl) was used as a negative control [45].

## 4. Conclusions 

A water-soluble, sulfated polysaccharide, F2-1, was obtained from the green alga, *C. isthmocladum*. F2-1 was found to be a branched heteropolysaccharide composed of Gal, Ara, Glc, Rha, and Xyl. The backbone was composed of a →4)-β-L-Arap(1→ residue and the 2-O and/or 3-O positions had sulfate modifications. Additionally, the 2-O or 3-O positions had branch points. The side chains were composed of →5)-β-D-Galf, (1→2,6)-β-D-Galf(1→, (1→2)-β-L-Rhap4S, →4)-α-D-Glcp(1→, and terminal α-D-Galp(1→ and β-D-Xylp(1→.

F2-1 exhibited mild anticoagulant activity on the intrinsic pathways of coagulation and thrombin activity. Although it does not have the same excellent anticoagulant activity as heparin, it may have a limited effect on the clotting pathway and will have fewer side effects and wider potential applications. Like most marine polysaccharides, F2-1 has a typical and complex structure. The fine structure of polysaccharides with the same resource in different seasons and regions was a little different. These factors will also limit the application of marine polysaccharides in the medical field. However, it may be more promising to study bioactive oligosaccharide fragments based on the same polysaccharides. Fucosylated chondroitin sulfates (FuCS), which are a unique type of polysaccharides in sea cucumber, show excellent anticoagulant activity. Their well-defined FuCS nonasaccharides consisting of a trisaccharide repeating unit of β-d-GalNAc(4,6-diS)-(1→4)-[α-l-Fuc(2,4-diS)-(1→3)]-β-d-GlcUA also display potent anticoagulant activity via the selective inhibition of the intrinsic tenase without bleeding risk [46]. The oligosaccharide is effective by oral administration compared with the intravenous injection of heparin. The approach to the synthesis of FuCS oligosaccharides was achieved [47]. Recently, the FuCS nonasaccharides successfully entered clinical trials as an anticoagulant. The research on F2-1 in the marine green alga, *Codium isthmocladum*, can also learn from this example. Further studies are required to demonstrate the anticoagulant mechanism of this compound. Therefore, *Codium isthmocladum* is worthy of further investigations as a candidate for functional food supplements or ingredients in the pharmaceutical industry. This would also serve as a useful method to solve the problem of algae blooms.

## Figures and Tables

**Figure 1 molecules-27-08012-f001:**
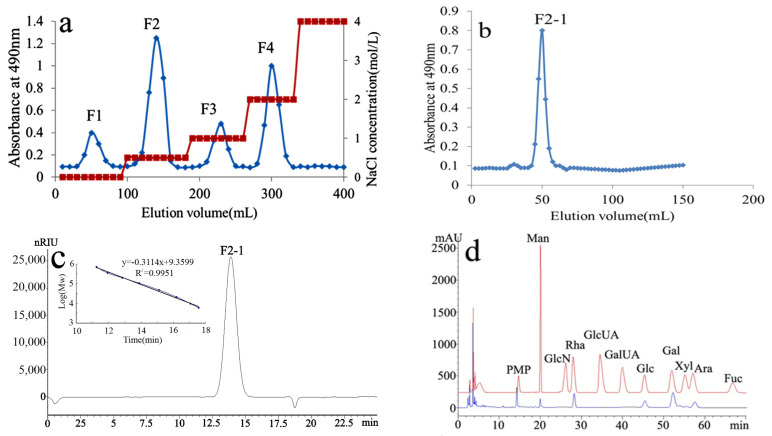
Purification, HPGPC, and HPLC chromatogram of the polysaccharide, F2-1, extracted from the green seaweed *Codium isthmocladum*. (**a**) Anion exchange chromatography using a Q Sepharose Fast Flow column was used to isolate the crude polysaccharide; (**b**) F2-1 was obtained using gel permeation chromatography of superdex 75; (**c**) HPGPC chromatogram of F2-1 using a TSKgel G3000PWXL column and standard curve of molecular weights; (**d**) monosaccharide composition of F2-1.

**Figure 2 molecules-27-08012-f002:**
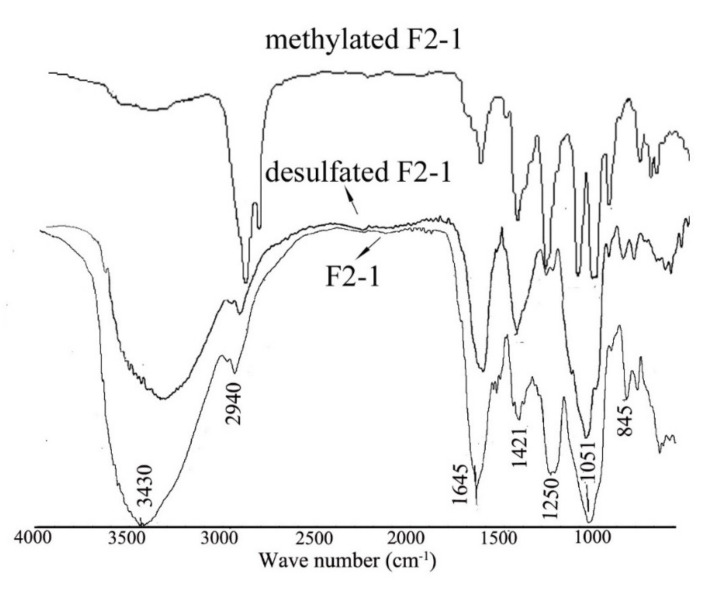
IR spectrum of F2-1, methylated F2-1, and desulfated F2-1.

**Figure 3 molecules-27-08012-f003:**
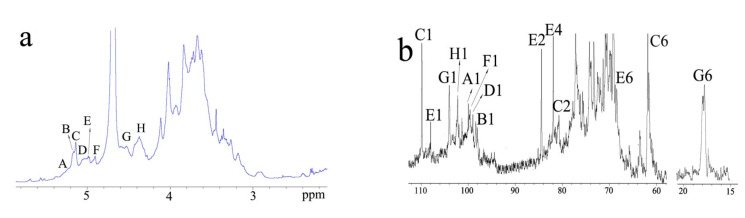
NMR spectra of F2-1. (**a**) ^1^H NMR spectrum; (**b**) ^13^C NMR spectrum; (**c**) part of ^1^H–^1^H COSY spectrum; (**d**) part of ^1^H–^1^H TOCSY spectrum; (**e**) ^1^H–^13^C HSQC spectrum.

**Figure 4 molecules-27-08012-f004:**
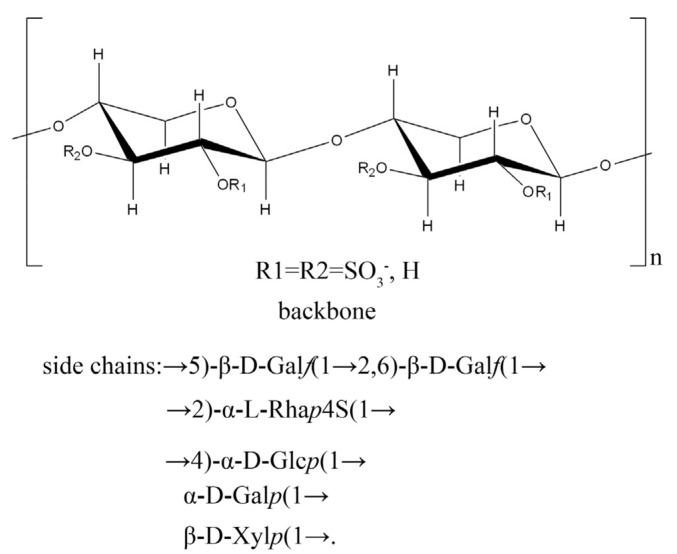
The possible structure of F2-1.

**Table 1 molecules-27-08012-t001:** GC–MS data analysis of partial *O*-methylated alditol acetates of F2-1.

Methylation Product	*m*/*z*	Molar Ratio (%)		Linkage Pattern
		F2-1	dsF2-1	
1,5-Di-*O*-acetyl-2,3,4-Tri-*O*-methyl-D-xyl	101, 117, 129, 161	4.8	7.9	xyl*p*(1→
1,5-Di-*O*-acetyl-2,3,4-Tri-*O*-methyl-L-Rha	101, 117, 131, 145, 161, 175	3.8	6.6	Rha*p*(1→
1,2,5-Tri-*O*-acetyl-3,4-di-*O*-methyl-L-Rha	115, 131, 189	4.8	19.6	→2)Rha*p*(→
1,4,5-Tri-*O*-acetyl-2,3-di-*O*-methyl-L-Ara	117, 129, 189	12.2	28.7	→4)Ara*p*(1→
1,2,3,4,5-Penta-*O*-acetyl- L-Ara	117, 127, 145, 159, 175, 187, 217	8.4	2.2	→2,3,4)Ara*p*(1→
1,5-Di-*O*-acetyl-2,3,4,6-Tetra-*O*-methyl-D-Man	101, 117, 129, 145, 161, 205	5.6	3.8	Man*p*(1→
1,5-Di-*O*-acetyl-2,3,4,6-Tetra-*O*-methyl-D-Glc	101, 117, 129, 145, 161, 205	5.4	3.9	Glc*p*(1→
1,5-Di-*O*-acetyl-2,3,4,6-Tetra-*O*-methyl-D-Gal	101, 117, 129, 145, 161, 205	7.3	8.9	Gal*p*(1→
1,2,4,5-Tetra-*O*-acetyl-3-*O*-methyl-L-Rha	129, 143, 189, 203	16.8	5.8	→2,4)Rha*p*(1→
1,4,5-Tri-*O*-acetyl-2,3-Di-*O*--methyl-D-Man	113, 117, 131,173, 233	6.8	7.3	→4)Glc*p*(1→
1,4,5-Tri-*O*-acetyl-2,3,6-Tri-*O*--methyl-D-Gal	113, 117, 131, 173, 233	10	3.1	→5)Gal*f*(1→
1,2,4,6-Tetra-*O*-acetyl-3,5-Di-*O*--methyl-D-Gal	117, 127, 129, 159, 189	17.1	6.2	→2,6)Gal*f*(1→

**Table 2 molecules-27-08012-t002:** The ^1^H and ^13^C NMR chemical shifts (δ) for the residues of F2-1.

Residue	H1/C1	H2/C2	H3/C3	H4/C4	H5/C5	H6/C6
A (1→4)-β-L-Ara*p*2S3S	5.31	4.7	4.5	—	—	—
	99	76	76.6	—	—	—
B (1→4)-α-D-Glc*p*	5.2	3.55	3.74	3.86	—	—
	97.4	73.2	72	76.8	—	—
C (1→5)-β-D-Gal*f*	5.16	4.18	3.9	4.08	3.78	3.68/3.75
	108.8	81.2	76	80.6	76.1	60.9
D (1→4)-β-L-Ara*p*	5.04	3.81	4.0	3.95	3.72	—
	98.3	68.2	68.4	75	—	—
E (1→2,6)-β-D-Gal*f*	5.0	4.12	4.16	3.91	3.8	3.57/3.68
	107	84.7	77	83.1	70.8	69
F terminal α-D-Gal*p*	4.92	3.83	3.95	3.76	—	—
	99	76.9	74.6	82.8	—	—
G (1→2)-β-L-Rha*p*4S	4.57	4.2	—	4.04	4.16	1.21
	103.3	78.2	—	66.7	72.6	17.6
H terminal β-D-Xyl*p*	4.4 102.1	3.26 73	3.46 72.6	3.57 —	— —	

**Table 3 molecules-27-08012-t003:** Anticoagulant activities of the sulfated polysaccharide F2-1 by APTT, TT, and PT compared with heparin.

Sample	Concentration (μg/mL)	APTT (S)	TT (S)	PT (S)
F2-1	0	35.8 ± 3.3	18.8 ± 2.1	13.6 ± 2.5
	2.5	55.7 ± 4.2	25.2 ± 2.9	16.5 ± 2.2
	5	87.5 ± 2.8	54.4 ± 3.0	16.9 ± 1.4
	10	133.0 ± 3.7	115.4 ± 2.2	17.6 ± 2.3
	20	>200	>120	17.6 ± 2.8
	50	>200	>120	19.9 ± 2.7
Heparin	0	35.8 ± 3.3	18.8 ± 2.1	13.6 ± 2.5
	2.5	88.9 ± 3.2	70.2 ± 3.2	46.6 ± 2.0
	5	118.3 ± 2.7	>120	59.7 ± 3.2
	10	>200	>120	68.8 ± 2.6
	20	>200	>120	89.1 ± 2.2
	50	>200	>120	>120

## Data Availability

The data presented in this study are available in the article.

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
