# Peer review of "Structure and Anticoagulant Activity of a Galactofuranose-Containing Sulfated Polysaccharide from the Green Seaweed, Codium isthmocladum"

_molecules, 2022, doi:10.3390/molecules27228012_

Round 1
Reviewer 1 Report
Heparin has widely served as anticoagulant agent due to its strong activities both in vivo and in vitro. However, there are some complications in its application,such as spontaneous bleeding. At present, some experimental studies have suggested that sulfate polysaccharides can exert an anticoagulant effect in vitro, but clinical evidence on the actual effectiveness is still rare. Research is relatively limited on the anticoagulant mechanism of natural polysaccharide, which might be related to the complexity in composition, structural diversity, in vivo low absorption and transport as macromolecules, and analysis difficulties caused by interference. Therefore, the reviewers think that it is of great scientific significance to explore the anticoagulant effect and mechanisms of polysaccharides. However, this article contains the following shortcomings, which need to be improved or corrected.
(1) It is superficial on content of the anticoagulant activities of F2-1. Basing on the current content of the article, evidence, indicated that F2-1 is a potential anticoagulant agent, is not thorough. We still don’t know advantages of it compare to existing anticoagulant agent. Additionally, it’s safety is still undefined.
(2) The characterization of F2-1 on the article is adequate. Nevertheless, it is necessary to use two methods to detect monosaccharide composition at least.
(3) The conclusion is already too simple and should be inserted the advantage and value of F2-1 polysaccharide as an anticoagulant agent to article. Types of anticoagulant agent are diverse, and they show various differences in the anticoagulant mechanism. We want to find out which situation are more apt being treated by the F2-1.
(4) Table 3 is not particularly visual. Whether these data can convert to line chart, or other graphic?
Author Response
Reviewer 1
1. It is superficial on content of the anticoagulant activities of F2-1. Basing on the current content of the article, evidence, indicated that F2-1 is a potential anticoagulant agent, is not thorough. We still don’t know advantages of it compare to existing anticoagulant agent. Additionally, it’s safety is still undefined.
Response: We feel great thanks for your professional review on this point. According to the current situation, heparin, low molecular weight heparin, heparin derivatives and minimal active unit of heparin are important anticoagulant drugs for clinical application. However, due to the multi-target and high biological activity of heparin, it still causes many adverse reactions, so it is still necessary to screen new anticoagulant drugs. Marine sulphated polysaccharides generally have anticoagulant activity, but due to the complexity of the structure, there is no successful marketing of Marine anticoagulant drugs. In 2019, an oligosaccharide of fucosylated Chondroitin Sulfate from sea cucumber which acts selectively on the endogenous clotting pathway and is effective orally was approval to enter clinical study by FDA. This example shows us that it is difficult to make marine polysaccharides into medicines, but the active fragments of the polysaccharides can still give many useful implications. In contrast, green algae polysaccharides have more significant anticoagulant activity than other marine polysaccharides.
Therefore, green algae polysaccharide can be used as a resource to study its active fragments and develop specific anticoagulant activity for anticoagulation and the treatment of thrombus related diseases. We think this is an important way to develop marine polysaccharides. In the text, we have revised some statements. Thank you again.
2. The characterization of F2-1 on the article is adequate. Nevertheless, it is necessary to use two methods to detect monosaccharide composition at least.
Response: PMP precolumn derivatization HPLC is a mature method for the determination of monosaccharide composition. Compared to high performance anion exchange chromatography (HPAEC) and GC, this method is convenient and can measure acid sugars, neutral sugars and amino sugars simultaneously. Involved in the different responsiveness of different monosaccharides in HPLC, we use the monosaccharide standards with the same mole ration at different concentrations to establish standard curve and achieve quantitative analysis. If needed, we could add the quantitative method of monosaccharide composition in the text.
3. The conclusion is already too simple and should be inserted the advantage and value of F2-1 polysaccharide as an anticoagulant agent to article. Types of anticoagulant agent are diverse, and they show various differences in the anticoagulant mechanism. We want to find out which situation are more apt being treated by the F2-1.
Response: We have revised the conclusion part as your suggestion, as seen in part“4. Conclusions”in revised version. Thank you!
4. Table 3 is not particularly visual. Whether these data can convert to line chart, or other graphic?
Response: Thank you for your advice. Because at higher concentrations, the blood coagulation
times of heparin and the sample are longer than the upper limit set. It is not very proper to use
line chart to present the data.
Reviewer 2 Report
The manuscript titled as “Structure and anticoagulant activity of a galactofuranosecontaining sulfated polysaccharide from the green seaweed, Codium isthmocladum” described that a water-soluble sulfated polysaccharide, F2-1, was obtained from the marine green alga. The detailed structural analysis was performed by methylation analysis and NMR to determine the monosaccharide composition and glycosidic linkages. F2-1 exhibited significant anticoagulant activity by prolonging the APTT and TT in vitro. The research is interesting but several important issues should be addressed before the manuscript was ready for publication.
1. Figure 1a: the unit of NaCl concentration is mg/mL, but the unit in the text is mol/L, please check and revise.
2. Line 73,H2SO4 should be H2SO4.
3. Figure 2: the stretching vibration of S=O is 1250 cm-1, but in the text it is 1249 cm-1, please check and revise.
4. Line 96: Except for bound water of the polysaccharide, the intensive band of 1646 cm–1 should also correspond to the C=O bond of uronic acids.
5. Line118-121: In Table 1 and line 118-121, the methylation product of 1,4,5-tri-O-acetyl-2,3,6-tri-O-methyl-D-Gal should include two linkage patterns: →4)Galp(1→ and →5)Galf(1→. So it was not proper to attribute it to furanosic galactose unit of →5)Galf(1→ only. According to the previously reported papers, the furanosic galactose units were rarely present in green seaweed of Codium isthmocladum. Please check further.
6. Figure 3: Lack of the 1H–13C HSQC spectrum.
7. The NMR data attribution of -α-L-Arap in Table 2 was not consistent with the previously reported literature. Please check and identified the right configuration for this polysaccharide with more solid evidences. Three papers are provided for reference:
([1] Han X W, Zhang Y Q, Liu L L, et al. Isolation, purification and physicochemical characteristics comparison study of polysaccharides between wild and low salinity cultured green seaweeds Chaetomorphalinum [J]. Chinese Journal of Marine Drugs, 2014. In this paper, the related signal H-1/C-1 (5.40/98.50 ppm) was assigned to →4)-β-L-Arap(2SO4)-(1→ residue.
[2] Qi X, Mao W, Gao Y, et al. Chemical characteristic of an anticoagulant-active sulfated polysaccharide from Enteromorpha clathrate [J]. Carbohydr Polym, 2012, 90(4): 1804-10. In this paper, the related signal H-1/C-1 (5.18/98.30 ppm) was assigned to →4)-β-L-Arap(3SO4)-1→ residue.
[3] He M, Hao J, Feng C, et al. Anti-diabetic activity of a sulfated galactoarabinan with unique structural characteristics from Cladophora oligoclada (Chlorophyta)[J]. Carbohydr Polym, 2022, 278: 118933.)In this paper, the related signal H-1/C-1 (5.29/98.7 ppm) was assigned to →3,4)-β-L-Arap-(1→ residue.
8. The “p” and “f ”in Table 2 and the whole manuscript should be italicized.
9. Line 257: the unit of NaCl concentration is mol/L not M. Check the whole manuscript.
10. Line 258-259: “The fractions were collected and the carbohydrates in each fraction……” should be expressed as: the fractions were collected and the content of the carbohydrates in each fraction was determined by phenol-sulfuric acid method.
11. Please check the references of this manuscript. There are errors and the format is not uniform.
Author Response
-Reviewer 2
1. Figure 1a: the unit of NaCl concentration is mg/mL, but the unit in the text is mol/L, please check and revise.
Response: Thank you for your kind and positive comments. We feel sorry for our carelessness. the unit of NaCl concentration in Figure 1a had been corrected to mol/L.
Line 73,H2SO4 should be H2SO4.
Response: We have corrected the errors. H2SO4 had been revised as H2SO4.
3.Figure 2: the stretching vibration of S=O is 1250 cm-1, but in the text it is 1249 cm-1, please check and revise.
Response: We feel sorry for our mistake. The stretching vibration of S=O in the text had been reviesed to 1250 cm-1 in revised version.
4. Line 96: Except for bound water of the polysaccharide, the intensive band of 1646 cm–1 should also correspond to the C=O bond of uronic acids.
Response: Thank you for your remind. We have revised this in the text(Line96).
5. Line118-121: In Table 1 and line 118-121, the methylation product of 1,4,5-tri-O-acetyl-2,3,6-tri-O-methyl-D-Gal should include two linkage patterns: →4)Galp(1→ and →5)Galf(1→. So it was not proper to attribute it to furanosic galactose unit of →5)Galf(1→ only. According to the previously reported papers, the furanosic galactose units were rarely present in green seaweed of Codium isthmocladum. Please check further.
Response: Thank you for your kind and positive comments. As you say, the PMAA of →4)Galp(1→ and →5)Galf(1→ have the same product 1,4,5-tri-O-acetyl-2,3,6-tri-O-methyl-D-Gal. Just from the methylation analysis, it is hard to confirm the linkage style. We were also confused by the data when we got the result. Because most galactofuranose-containing polysaccharides are found in microorganism. But many evidences indicated the polysaccharide in Codium isthmocladum contained galactofuranose. First of all, beside →5)Galf(1→, we also found the product of →2,6)Galf(1→ with big amounts. Through research literature, we found arabinose could also exisit in algae as furan configuration. But the PMAA data of Araf is different from that of Galf in peak times and mass spectrum fragments. And the NMR spectrum also confirmed the characteristics of β-D-Galf. It is not very common but was found in marine green algae. In our opinion, to confirm the possibility of the existence of Galf, the genes associated with galactofuranose synthesis should be detected. If needed, we could provide the original data on methlylation and GC-MS.
6. Figure 3: Lack of the 1H–13C HSQC spectrum.
Response: Thank you for your remind. We have added the HSQC spectrum in the text(Line204-207).
7. The NMR data attribution of -α-L-Arap in Table 2 was not consistent with the previously reported literature. Please check and identified the right configuration for this polysaccharide with more solid evidences. Three papers are provided for reference:
([1] Han X W, Zhang Y Q, Liu L L, et al. Isolation, purification and physicochemical characteristics comparison study of polysaccharides between wild and low salinity cultured green seaweeds Chaetomorphalinum [J]. Chinese Journal of Marine Drugs, 2014. In this paper, the related signal H-1/C-1 (5.40/98.50 ppm) was assigned to →4)-β-L-Arap(2SO4)-(1→ residue.
[2] Qi X, Mao W, Gao Y, et al. Chemical characteristic of an anticoagulant-active sulfated polysaccharide from Enteromorpha clathrate [J]. Carbohydr Polym, 2012, 90(4): 1804-10. In this paper, the related signal H-1/C-1 (5.18/98.30 ppm) was assigned to →4)-β-L-Arap(3SO4)-1→ residue.
[3] He M, Hao J, Feng C, et al. Anti-diabetic activity of a sulfated galactoarabinan with unique structural characteristics from Cladophora oligoclada (Chlorophyta)[J]. Carbohydr Polym, 2022, 278: 118933.)In this paper, the related signal H-1/C-1 (5.29/98.7 ppm) was assigned to →3,4)-β-L-Arap-(1→ residue.
Response: Thank you to provide proper references for us. We are sorry to make such an obvious mistake. In the text, we confirm the Ara residues were β-L-Arap. But in the table, we made the mistake. We have checked and revised them. We will read and check the paper more carefully in the future. Thank you for your careful review. We have added your suggested references in the proper places in the text.
8. The “p” and “f ”in Table 2 and the whole manuscript should be italicized.
Response: Thank you for your kind and positive comments. We feel sorry for our carelessness. The “p” and “f ” in all the tables and whole manuscript had been italicized.
9.Line 257: the unit of NaCl concentration is mol/L not M. Check the whole manuscript.
Response: We have checked the whole manuscript and revised this errors.
10. Figure 2: Line 258-259: “The fractions were collected and the carbohydrates in each fraction……” should be expressed as: the fractions were collected and the content of the carbohydrates in each fraction was determined by phenol-sulfuric acid method.
Response: That is revised. Thank you!
11. Please check the references of this manuscript. There are errors and the format is not uniform.
Response: Thank you for your kind reminder and comments. We have checked the references of this manuscript and the format of all references have been unified according to the format requirements of the journal, and three references you mentioned aboved were also added.
Thank you again for your positive comments and valuable suggestions to improve the quality of our manuscript. We tried our best to improve the manuscript and these changes will not influence the content and framework of the paper.
Round 2
Reviewer 1 Report
There were no comments or suggestions for authors basing on the revised manuscript and the aurthors‘ responce.
Author Response
Thank you for your comments.
Reviewer 2 Report
1. Line 3: In the abstract, “the backbone of F2-1 was →4)-α-L-Arap(1→ residue” should be “the backbone of F2-1 was →4)-β-L-Arap(1→ residue”.
2. Line 96: For the sentence of “Excluding the existence of the C=O bond of uronic acids”, the use of word “Excluding” was not proper here. Excluding means not including. Please check and select proper words.
3. Please Check the whole manuscript carefully to avoid the above errors
Author Response
Response for editors and reviewers:
-Reviewer 1
1. Line 3: In the abstract, “the backbone of F2-1 was →4)-α-L-Arap(1→ residue” should be “the backbone of F2-1 was →4)-β-L-Arap(1→ residue”.
Answer: We are ashamed to miss the revise of abstract. Thank you again for your careful review.
- Line 96: For the sentence of “Excluding the existence of the C=O bond of uronic acids”, the use of word “Excluding” was not proper here. Excluding means not including. Please check and select proper words.
Answer: We have revised the misunderstanding description. Thank you!
- Please Check the whole manuscript carefully to avoid the above errors
Answer: We have revised the whole text. The revised places are in red.
Thank you again for your positive comments and valuable suggestions to improve the quality of our manuscript. We tried our best to improve the manuscript and these changes will not influence the content and framework of the paper.